# Tumor Resection in Hepatic Carcinomas Restores Circulating T Regulatory Cells

**DOI:** 10.3390/jcm13196011

**Published:** 2024-10-09

**Authors:** Carmen Martín-Sierra, Ricardo Martins, Margarida Coucelo, Ana Margarida Abrantes, Rui Caetano Oliveira, José Guilherme Tralhão, Maria Filomena Botelho, Emanuel Furtado, Maria Rosário Domingues, Artur Paiva, Paula Laranjeira

**Affiliations:** 1Flow Cytometry Unit, Department of Clinical Pathology, Hospitais da Universidade de Coimbra, Unidade Local de Saúde de Coimbra, 3000-076 Coimbra, Portugal; carmenms.btg@gmail.com; 2Coimbra Institute for Clinical and Biomedical Research (iCBR), Center of Environmental Genetics of Oncobiology (CIMAGO), Faculty of Medicine (FMUC), University of Coimbra, 3000-548 Coimbra, Portugal; ricardo.martins@ulscoimbra.min-saude.pt (R.M.); margarida.coucelo@ulscoimbra.min-saude.pt (M.C.); mabrantes@fmed.uc.pt (A.M.A.); rui.caetano@cedap.pt (R.C.O.); jgtralhao@ulscoimbra.min-saude.pt (J.G.T.); mfbotelho@fmed.uc.pt (M.F.B.); 3Center for Innovative Biomedicine and Biotechnology (CIBB), University of Coimbra, 3004-504 Coimbra, Portugal; 4Clinical Academic Center of Coimbra (CACC), 3000-061 Coimbra, Portugal; 5Unidade Transplantação Hepática Pediátrica e de Adultos, Centro Hospitalar e Universitário de Coimbra, 3000-075 Coimbra, Portugal; emanuelfurtado@ulscoimbra.min-saude.pt; 6Serviço de Cirurgia Geral, Unidade HBP, Centro Hospitalar e Universitário de Coimbra, 3000-075 Coimbra, Portugal; 7University of Coimbra, Faculty of Medicine, Biophysics Institute, 3000-548 Coimbra, Portugal; 8Unidade Funcional de Hematologia Molecular, Serviço de Hematologia Clínica, Centro Hospitalar e Universitário de Coimbra, 3000-075 Coimbra, Portugal; 9Serviço de Anatomia Patológica, Centro Hospitalar e Universitário de Coimbra, 3000-075 Coimbra, Portugal; 10CESAM—Centre for Environmental and Marine Studies, Department of Chemistry, University of Aveiro, 3810-193 Aveiro, Portugal; mrd@ua.pt; 11Mass Spectrometry Center, LAQV-REQUIMTE, Department of Chemistry, University of Aveiro, 3810-193 Aveiro, Portugal; 12Instituto Politécnico de Coimbra, ESTESC-Coimbra Health School, Ciências Biomédicas Laboratoriais, 3046-854 Coimbra, Portugal; 13Center for Neuroscience and Cell Biology (CNC), University of Coimbra, 3004-504 Coimbra, Portugal

**Keywords:** hepatocellular carcinoma, cholangiocarcinoma, primary liver cancer, T cells, Treg, NK, IFN-γ, IL-17, TGF-β, IL-10

## Abstract

**Background/Objectives**: Cholangiocarcinoma (CCA) and hepatocellular carcinoma (HCC) represent major primary liver cancers, affecting one of the most vital organs in the human body. T regulatory (Treg) cells play an important role in liver cancers through the immunosuppression of antitumor immune responses. The current study focuses on the characterization of circulating natural killer (NK) cells and T cell subsets, including Treg cells, in CCA and HCC patients, before and after surgical tumor resection, in order to understand the effect of tumor resection on the homeostasis of peripheral blood NK cells and T cells. **Methods**: Whole blood assays were performed to monitor immune alterations and the functional competence of circulating lymphocytes in a group of ten healthy individuals, eight CCA patients, and twenty HCC patients, before and one month after the surgical procedure, using flow cytometry, cell sorting, and qRT-PCR. **Results**: Before tumor resection, both HCC and CCA patients display increased percentages of CD8^+^ Treg cells and decreased frequencies of circulating CD4^+^ Treg cells. Notwithstanding, no functional impairment was detected on circulating CD4^+^ Treg cells, neither in CCA nor in HCC patients. Interestingly, the frequency of peripheral CD4^+^ Treg cells increased from 0.55% ± 0.49 and 0.71% ± 0.54 (in CCA and HCC, respectively) at T0 to 0.99% ± 0.91 and 1.17% ± 0.33 (in CCA and HCC, respectively) at T1, following tumor resection. **Conclusions**: Our results suggest mechanisms of immune modulation induced by tumor resection.

## 1. Introduction

The liver, one of the most vital organs in the human body, is considered to be an immune-privileged organ. Despite that, it shelters a multitude of immunocompetent cells (T cells, B cells, natural killer [NK] cells, and dendritic cells) that are able to respond promptly when challenged with immunogenic antigens [1]. Hepatocellular carcinoma (HCC) is the most frequent liver cancer, one of the most prevalent cancers worldwide, and is considered to be the second leading cause of cancer-related death worldwide [2,3]. Its incidence rate shows great geographical variation, being approximately 4/100,000 in Europe [4,5]. Cholangiocarcinoma (CCA) is the second most common liver cancer. It originates from biliary epithelial cells and is characterized by an aggressive clinical course, associated with a high mortality rate [6,7]. The incidence rate of CCA also varies worldwide, with an annual incidence ranging from 0.4/100,000 to 1.8/100,000 in Europe, and from 0.6/100,000 to 1.0/100,000 in the United States [8].

The only effective treatments for patients with CCA or HCC are tumor resection or liver transplantation; thus, novel therapeutic approaches, such as immunotherapies, are required [9]. The tumor-mediated immunosuppression, or the direct suppression of effector immune cells, is one of the main mechanisms proposed for an attenuated immune response against tumors [10]. There are several immune cell populations displaying immunosuppressive activity, but T regulatory (Treg) cells assume a prominent role in hindering an effective antitumor response. In fact, the amount of Treg cells infiltrating the tumor was shown to influence tumor progression and prognosis in some cancers [11]. Accordingly, Treg cells became a potential target for the treatment of malignant diseases, and the search for the best molecules to hamper Treg function, or deplete Treg cells, is an active field of research, with several ongoing clinical trials [11,12]. Interestingly, in individuals infected with HIV, the levels of circulating Treg cells were found to be negatively correlated with tumor development [13].

The local lymphocyte population infiltrating the liver is enriched in NK and NKT cells, with a possible critical role in the recruitment of circulating T cells [14]. A marked infiltration of T regulatory (Treg) cells, with the ability to suppress the inflammatory response resulting from innate and adaptive immunity, has been observed in the livers of patients with HCC, and the number of intratumoral Treg cells is increased compared to the peritumoral regions and the periphery [1]. CD4^+^ Treg cells are a subgroup of CD4^+^ T cells, characterized by the expression of FoxP3, the expression of the interleukin (IL)-2 receptor α chain (CD25) at high levels, and the low expression or absence of the alpha chain of the IL-7 receptor (CD127) [15]. There is accumulating clinical evidence that CD4^+^ Treg cells, together with other immune-regulatory cell subsets, play an important role in liver cancers, such as HCC, through the inhibition of protective non-specific and tumor-specific immune responses [16]. Considering the importance of CD4^+^ Treg cells in inhibiting an effective antitumor immune response, we have quantified and investigated the function of this cell subset in the peripheral blood (PB) of CCA and HCC patients before tumor resection (T0) and once the patients were recovered from surgery (T1). In parallel, CD8^+^ Treg cells and other T cell subsets were studied. A group of age- and sex-matched healthy adults (HG) was also studied in order to understand the immune status of the different T cell populations analyzed, and to obtain evidence about the changes in the proportions and/or function of distinct T cell subsets after tumor resection. Monitoring the peripheral immune response after surgical resection will provide information to advance our understanding about the mechanisms underlying the clinical response to surgical resection in these carcinomas. Here, we found that CCA and HCC patients display decreased levels of CD4^+^ Treg circulating in the blood, which are restored one month after the surgical resection of the tumor.

## 2. Materials and Methods

### 2.1. Patients and Healthy Individuals

A total of 20 patients with HCC (3 women and 17 men; average age: 62.2 ± 14.5 years) and 8 patients with CCA (5 women and 3 men; average age: 61.0 ± 14.7 years) were included in this study. PB samples were collected at the time point of the surgical resection, just before the beginning of the surgical intervention (T0), and once the patient was completely recovered from the surgery (T1), corresponding to a timepoint ranging from 1 to 6 months, and generally 1 to 2 months after surgery. A group of 10 age- and sex-matched healthy individuals was included in the study as a control group (HG). None of the patients took any antitumor therapy or medication prior to surgery, nor at T1. Nonetheless, 7 HCC patients underwent liver transplantation, and tacrolimus was administrated to them just after the surgical procedure. Tacrolimus targets T lymphocytes [17,18]. Accordingly, HCC patients who underwent liver transplantation were excluded from the comprehensive analysis at T1 to ensure that immunosuppressive effects of post-transplant drugs, like tacrolimus, did not interfere with the natural restoration of Treg cells. The clinical information about the patients included in this study is detailed in Table 1.

The study protocol was approved by the Ethical Committee from the Faculty of Medicine, University of Coimbra, Coimbra, Portugal (CE-136/2016). Informed consent was obtained from all individual participants included in the study.

### 2.2. Phenotypic Characterization of Circulating T Cells and NK Cells

Peripheral blood (PB) samples were collected from participants and healthy individuals into K3-EDTA and heparin tubes. Blood samples were stained using the monoclonal antibody (mAb) combinations detailed in Table 2. For the staining of surface antigens, as described for tube 1 (Table 2), 300 μL of PB were incubated with the mAbs for 10 min in the dark, at room temperature (RT). Then, samples were incubated with 2 mL of FACS Lysing solution (Becton Dickinson Biosciences (BD), San Jose, CA, USA) for 10 min in the dark, at RT, and then centrifuged for 5 min at 540 g. The supernatant was discarded, and the cell pellet was washed in 1 mL of PBS (BD), resuspended in 500 μL of PBS, and immediately acquired in the flow cytometer.

To evaluate cytokine production by PB T cells and NK cells, lymphocytes were stimulated with phorbol myristate acetate (PMA: 0.25 μg/mL, Sigma-Aldrich, Burlington, MA, USA) and ionomycin (1 μg/mL, Boehringer Mannheim, Mannheim, Germany), which were added to 250 µL of heparin-collected PB, previously diluted as 1:1 (*v*/*v*) in RPMI 1640 complete culture medium (Invitrogen, Waltham, MA, USA). Brefeldin-A (10 µg/mL, Sigma-Aldrich) was also used to prevent the release of the newly synthesized cytokines. All samples were then incubated in a sterile environment with humid atmosphere and 5% CO_2_, at 37 °C, for 4 h.

Immunophenotypic analysis of circulating T and NK cells, cultured in the presence of PMA plus ionomycin, was performed by using a seven-color mAbs combination, detailed in Table 2 (tube 2). Cells were stained with the mAbs for surface proteins antigens (CD45, CD4, CD56, CD3, and CD8) and, after an incubation period of 10 min at RT in the dark, washed with PBS. For intracellular staining, the Fix&Perm (Life Technologies, Austin, TX, USA) reagent was used, in accordance with the instructions of the manufacturer, and samples were stained with the mAbs for IFN-γ and IL-17 (Table 2, tube 2). After washing twice with PBS, the cell pellet was resuspended in 500 μL of PBS, and immediately acquired in the flow cytometer.

### 2.3. Flow Cytometry Data Acquisition and Analysis

Data acquisition was performed in a FACSCanto II flow cytometer (BD), and analyzed using Infinicyt 1.8 software (Cytognos SL, Salamanca, Spain).

T cells were identified based on CD3 and CD45 positivity and intermediate FSC and SSC properties. Within this cell population, CD4^+^ (Th), CD8^+^ (Tc), and CD4^+^CD8^+^ T cell subsets (phenotypically characterized as CD3^+^CD4^+^CD8^−^, CD3^+^CD4^−^CD8^+^, and CD3^+^CD4^+^CD8^+^, respectively) were identified. CD4^+^ Tregs were further identified within CD4^+^ T cells, according to their typical phenotype of CD3^+^CD4^+^CD25^++^CD127^low/−^. Likewise, CD8^+^ Tregs were identified by the CD3^+^CD8^+^CD25^++^CD127^low/−^ phenotype. Among Th and Tc cells, we were able to identify Th1 and Tc1 (CD4^+^ T and CD8^+^ T cells, respectively, expressing IFN-γ), Th17 and Tc17 (CD4^+^ T and CD8^+^ T cells, respectively, expressing IL-17), and Th1/17 and Tc1/17 (CD4^+^ T and CD8^+^ T cells, respectively, simultaneously expressing IFN-γ and IL-17). Fluorescence minus one controls were used to confirm the gate establishment, particularly for the identification of cytokine-producing T cells. NK cells were identified on the basis of CD45 and CD56 positivity, absence of CD3 expression, and intermediate FSC and SSC properties.

### 2.4. Cell Purification by Fluorescence-Activated Cell Sorting

PB Treg cells were purified by fluorescence-activated cell sorting (FACS), using a FACSAria III flow cytometer (BD), according to its typical phenotype. Thus, the six-color mAbs combination we used (Table 2, tube 1) allowed the identification of CD4^+^ Tregs (CD3^+^CD4^+^CD25^++^CD127^low/−^), as depicted in Figure 1A.

For subsequent mRNA expression analysis, purified Treg cells were centrifuged for 5 min at 300× *g*, the supernatant was discarded, and the cell pellet was resuspended in 350 μL of RLT Lysis Buffer (Qiagen, Hilden, Germany), and stored at −80 °C.

### 2.5. RNA Isolation and Quantitative Real-Time Reverse Transcriptase-Polymerase Chain Reaction

The RNeasy™Micro Kit (Qiagen) was used to extract and purify total RNA, according to the manufacturer’s instructions. Reverse transcription was performed using a SensiScript Reverse Transcription Kit (Qiagen) according to the supplier’s instructions, and with Random Hexamer Primer (Thermo Fisher Scientific, San Jose, CA, USA). Relative quantification of gene expression was performed in a QuantStudio (Thermo Fisher Scientific) by a quantitative real-time reverse transcriptase-polymerase chain reaction (qRT-PCR). qRT-PCR was conducted using the PowerUp™ SYBR™ Green Master Mix (Thermo Fisher Scientific), using optimized primers for TGFβ, FOXP3, IL-10, and endogenous control glyceraldehyde 3-phosphate dehydrogenase (GAPDH) (Qiagen), according to the manufacturer’s instructions.

### 2.6. Statistical Analysis

For all of the studied variables, their mean values, standard deviation, median, and range were calculated. Due to the small sample size, the non-parametric Mann–Whitney and Kruskal–Wallis multiple comparison tests were performed to determine the statistical significance of the differences observed between groups. The non-parametric Wilcoxon test was performed to compare T1 vs. T0. Statistical tests were carried out using the Statistical Package for Social Sciences software (SPSS, version 25, IBM, Armonk, NY, USA). Statistically significant differences were considered when *p* < 0.05. GraphPad Prism software (version 8.0.1) was used to create the graphics.

## 3. Results

### 3.1. CCA and HCC Patients Display a Reduction of Circulating CD4^+^ Treg Cells That Is Restored after Surgical Tumor Resection

Figure 1B demonstrates a significant reduction in circulating CD4^+^ Treg levels (measured in whole blood) in both CCA and HCC patients, at T0, compared to healthy controls (*p* < 0.05), with levels recovering post-surgery (T1). This suggests a potential migration of Tregs into the tumor microenvironment pre-surgery. Mean values and standard deviation for these parameters are detailed in Appendix A. Likewise, the absolute numbers of peripheral blood CD4^+^ Treg cells are significantly decreased for CCA (*p* < 0.05) and HCC (*p* > 0.05) patients at T0, as compared to the HG. After tumor resection (T1), a recovery in the absolute counts of CD4^+^ Treg cells from CCA and HCC patients (Figure 1B and Appendix A) is observed, suggesting that, after tumor removal, the active migration of Treg cells to the liver ceases, and the Treg cell population in the peripheral blood is replenished. Conversely, at T0, both CCA and HCC patients show a tendency to have an increased percentage of CD8^+^ Treg cells (measured within CD8^+^ T cells), corresponding to 0.25 ± 0.22 and 0.37 ± 0.48, respectively (vs. 0.16 ± 0.21 in HG); and augmented CD8^+^ Tregs counts (CCA: 0.69 ± 0.83, HCC: 1.18 ± 1.17, HG: 0.59 ± 0.25), not reaching statistical significance (Appendix A). Interestingly, at T1, CCA patients present percentages of CD8^+^ Tregs (within CD8^+^ T cells) that are similar to those observed in the HG (Appendix A). Though CD8^+^ Tregs are a less-represented cell population (scarcer than CD4^+^ Tregs), these cells possess potent immunosuppressive activity. The increased number of circulating CD8^+^ Tregs, observed at T0, can be a compensatory mechanism in response to the decrease in CD4^+^ Tregs. On the other hand, the increase observed in the percentage of CD8^+^ Tregs (measured within CD8^+^ T cells) may be a consequence of the increased absolute number of CD8^+^ Tregs, plus the active migration of (non-regulatory) CD8^+^ T cells into the tumor site. This would explain the re-establishment of the CD8^+^ Tregs’ percentage (measured within CD8^+^ T cells) at T1.

Regarding the functional characterization of CD4^+^ Tregs, no significant differences were observed among the groups under study for IL-10, TGFβ, and FoxP3 mRNA expression (Figure 1C), indicating that CD4^+^ Tregs from CCA and HCC have no functional differences from HG CD4^+^ Treg cells.

### 3.2. Alterations in CD4^+^ Tregs and CD8^+^ Tregs Are Detected When Considering Subgroups of CCA and HCC Patients

Remarkably, after grouping the CCA and HCC patients according to the TNM stage, histologic grade, absence/presence of cirrhosis, absence/presence of microvascular invasion, and disease aggressiveness, some differences in the percentage of Treg cells, measured at T0, were revealed (Figure 2, Table 3 and Appendix A), demonstrating that the clinical parameters of patients, and the biological features of the tumor, impact the immune system at the peripheral level. In CCA, the percentage of CD4^+^ Tregs presents a tendency to increase from TNM stage II to stage IV, while the percentage of CD8^+^ Tregs decreases (Figure 2, Table 3 and Appendix A). In turn, the percentage of CD4^+^ Tregs tends to sequentially decrease from histologic grade G1 to G2 and G3 (Figure 2, Table 3 and Appendix A). Finally, HCC patients without cirrhosis present levels of CD8^+^ Tregs that are similar to those found in the control group (HG), contrasting with the HCC cirrhotic patients who present double of the percentage of CD8^+^ Tregs (Figure 2, Table 3 and Appendix A). This finding points to a possible role of liver cirrhosis in the expansion of the CD8^+^ Treg cell population. The presence of microvascular invasion and the aggressiveness degree seem not to impact the levels of peripheral Treg cells (Figure 2, Table 3, and Appendix A).

Notably, despite the differences found among CCA and HCC subgroups, our data show that there is a general tendency for both CD4^+^ Tregs and CD8^+^ Tregs to approach the levels observed in the HG, at T1, regardless of the subgroup (Appendix A). Remarkably, one exception is observed for HCC cirrhotic patients, whose levels of CD8^+^ Tregs remain increased after tumor resection (Appendix A), further supporting the possible participation of liver cirrhosis in CD8^+^ Treg expansion. Appendix A contains a detailed description of CD4^+^ Treg and CD8^+^ Treg levels (in percentage and absolute counts) in the peripheral blood of HG, CCA, and HCC (both at T0 and T1), and patients’ subgroups (grouped according to the TNM stage, histologic grade, negativity/positivity for the hepatitis C virus, absence/presence of cirrhosis and microvascular invasion, and aggressiveness degree of the tumor).

After a follow-up period equal to or longer than 12 months (ranging from 12 to 24 months), three out of seven CCA patients relapsed and died due to CCA. Those individuals tended to present lower percentages and absolute numbers of CD4^+^ Tregs at T0, which did not recover after tumor resection, while in relapse-free individuals, the lower levels of CD4^+^ Treg cells observed at T0 achieved normal levels at T1, as described in Table 4. Interestingly, CD8^+^ Treg cells also tended to be at lower levels, at T0, in patients with recurrent CCA, as compared to recurrence-free patients and healthy individuals (Table 4). During the follow-up period, none of the HCC patients relapsed.

### 3.3. Alterations of Other T Cell Subsets and NK Cells in CCA and HCC Patients

Both CCA and HCC patients tend to display decreased percentages of PB lymphocytes and T cells (*p* > 0.05), accompanied by the reduction of the absolute number of lymphocytes circulating in the PB (*p* > 0.05), as described in Table 5.

Regarding T lymphocyte subsets, both CCA and HCC patients show a decreased percentage of IFNγ^+^IL-17^+^ CD4 T cells (Th1/17), IL-17^+^ CD4 T cells (Th17), and IL-17^+^ CD8 T cells (Tc17). This decrease reaches statistical significance, when compared to the HG, for IFNγ^+^IL-17^+^ CD4 T cells from CCA patients at T0, and for IL-17^+^ CD8 T cells in CCA patients, both at T0 and T1 (Table 5).

Finally, regarding NK characterization, an increased frequency of peripheral CD56^BRIGHT^CD8^+^ NK cells in both CCA and HCC patients has been detected, only reaching statistical significance for HCC patients at T1 (Table 6).

## 4. Discussion

The current study focuses on the characterization of peripheral T cell subsets, more specifically Treg cells, in CCA and HCC patients, before (T0) and after (T1) surgical resection of the tumor, in order to understand the immune status of these patients and assess the effect of tumor resection on the T cell homeostasis.

Despite the great importance Treg cells have in the oncology field, where they are increasingly recognized for their impact on the disease’s aggressiveness and prognosis, there are a limited number of publications focusing on these cells in liver cancers. To the best of our knowledge, this is the first study reporting CD8^+^ Treg cells in CCA and HCC. The alterations observed in circulating CD8^+^ Tregs from CCA and HCC patients indicate a potential active role of these cells in the physiopathology of these cancers.

We found that both CCA and HCC patients displayed a significant decrease in the percentage of circulating CD4^+^ Treg cells before the surgical removal of the tumor, as compared to the HG, with a partial recovery in some CCA patients, and a total recovery in HCC patients one month after tumor resection. Likewise, absolute numbers of peripheral blood CD4^+^ Treg cells were decreased in HCC and CCA patients before surgery, and significantly increased after tumor resection. This increase is more marked in HCC patients, approaching the values detected in the HG. Previous studies of HCC found increased levels of CD4^+^ regulatory T cells in the blood (prior to surgery) [19,20,21]. However, this apparent discrepancy can be explained by the different approach used for the identification of Treg cells, since, in two of the three abovementioned studies [19,20], all CD4^+^CD25^+^ T cells were considered Tregs. However, the CD4^+^CD25^+^ T cell subset includes both Treg cells and activated T cells. Treg cells, which constitute a minority of the CD4^+^CD25^+^ T cell subset, present low expression levels or an absence of CD127, along with high expression levels of CD25 and FoxP3 (this strategy was used to identify Tregs in our study). The remaining CD4^+^CD25^+^ T cell subset corresponds to activated T cells, which are not Treg cells, and constitute the majority of the CD4^+^CD25^+^ T cell subset. The observed decrease in circulating CD4^+^ Treg cells in CCA and HCC patients, at T0, observed here, indicates a possible migration of this T cell population into the tumor microenvironment. This is further supported by the increase in the circulating levels of CCL20 observed in this cohort of patients, in our previous study [22], and by a number of studies that have detected infiltration of Treg cells in the tumor microenvironment of HCC and CCA patients [23,24,25,26]. In the same line, higher levels of Tregs were found in HCC patients’ livers in comparison to livers of healthy individuals [27]. In this regard, among the intratumoral Treg cells there is a high proportion of cells with enhanced immunosuppressive potential, compared to peritumoral tissue; therefore, Treg cells can become a promising prognostic factor in patients with hepatic carcinomas [26,28]. A recent study on a small cohort of CCA patients (n = 6) reported that higher levels of CD4^+^ Treg cells, either in circulation or infiltrating the tumor, were associated with a longer relapse-free survival [29], though this finding is controversial [26]. This contradictory finding may be explained by the limited sample size, or one can postulate that perhaps the functional status of Treg cells from liver cancer patients may display interindividual variability. Indeed, a study carried out by Alvisi and colleagues [26] on a larger cohort of CCA patients (n = 147) demonstrated that a dominant presence of CD4^+^ Treg cells overexpressing MEOX1 in the tumor infiltrate was associated with a poorer prognosis and shorter overall survival, as compared to individuals presenting lower MEOX1 expression. MEOX1 is a transcription factor which, when expressed at high levels by CD4^+^ Treg cells, endows them with increased immunosuppressive activity. In turn, in HCC patients, higher levels of tumor-infiltrating Tregs have always been associated with a poorer prognosis [28,30,31,32].

To the best of our knowledge, there are no published reports evaluating the effect of tumor resection on circulating Treg cells. Here, we first detected the recovery of CD4^+^ Treg cells in circulation after surgical resection on CCA and HCC patients, suggesting restoration of immune system balance following tumor resection. We may hypothesize that the increase on circulating CD4^+^ Treg, detected after tumor resection, may have occurred because the active migration of Treg cells into the liver ceased once the tumor was removed; therefore, Treg cells accumulated in the peripheral blood until they achieved homeostatic levels, similar to those observed in healthy individuals. Given the crucial role of Treg cells in maintaining self-tolerance and regulating the inflammatory response, as well as their potential to shape antitumor immune response, the restoration of Treg cells’ numbers and function is of upmost importance for the immune homeostasis.

In addition, we used qRT-PCR to analyze the mRNA expression of FoxP3, IL-10, and TGFβ in purified blood CD4^+^ Tregs from CCA and HCC patients, as well as from healthy donors. FoxP3 is an important transcription factor essential for immunosuppressive Treg differentiation and function [11,33]. We found high levels of FoxP3 mRNA in CD4^+^ Tregs purified by cell sorting, and no differences were detected among the biological groups under study, confirming the T regulatory phenotype of the purified cell populations. Likewise, no significant alterations were detected when evaluating circulating CD4^+^ Treg cells’ function by IL-10 and TGFβ mRNA expression, indicating that there are no functional defects in this cell subset in HCC or CCA patients. IL-10 and TGFβ are important immune-regulatory cytokines for Treg activity. In fact, Treg cells can exert their immunosuppressive function by diverse mechanisms, including contact-dependent mechanisms (mediated by proteins expressed at the surface of the Treg cell, such as PD-L1, CD39, ICOS, GITR, CTLA-4, and CD200), immunomodulatory cytokines’ secretion (namely, IL-10 and TGF-β), as well as through metabolic perturbation of target cells [34]. Interestingly, a study performed by Alvisi and colleagues [26] demonstrated an enhanced immunosuppressive potential of CCA-infiltrating CD4^+^ Treg cells when compared to their peritumoral or blood counterparts, associated with an increased expression of the immune-suppressive molecules PD-1 (programmed cell death 1 or CD279), PD-L1 (programmed death-ligand 1 or CD274), ICOS (inducible T cell co-stimulator or CD278), GITR (glucocorticoid-induced tumor necrosis factor receptor-related protein or CD357), and CD39, indicating that an improvement of the CD4^+^ Tregs’ immunomodulatory abilities may be induced by the tumor microenvironment. From this perspective, no functional differences between circulating Treg cells from CCA and healthy individuals would be expected, which is in accordance with our results.

Notably, with the demonstration that CD4^+^ Treg cells infiltrating liver tumors express a panoply of immune-suppressive molecules [26,35], it becomes clear that tumor-infiltrating Tregs dispose of a multitude of mechanisms to hinder the antitumoral activity of the different types of immune cells present in the tumor microenvironment. Accordingly, increased expression of PD-L1 in HCC predicts poor prognosis [36]. Furthermore, CD200, present at the plasma membrane of CD4^+^ Treg cells, was found to interact with CD200R1 from myeloid cells in the CCA microenvironment [26]. Animal models and in vitro models with human cells demonstrated that CD200R–CD200 interaction prevents Ras, MAPK, and NF-kB activity in CD200R-bearing cells, resulting in the reduction of several proinflammatory cytokines’ expression (namely, TNF-α, IFNγ, IL-1, IL-6, and IL-17), accompanied by the upregulation of the anti-inflammatory molecules IL-10 and TGF-β, in myeloid cells, along with the induction of tolerogenic dendritic cells [37,38,39]. On T cells, CD200 signaling promotes their differentiation into Treg or Tr1 cells expressing IL-10 and TGF-β [37]. A detailed description of the mechanisms and action of immune-suppressive molecules on different types of immune cells can be found in a previous publication [40].

To the best of our knowledge, this is the first study reporting CD8^+^ Treg cells in CCA and HCC. The frequency and absolute numbers of these cells are increased in CCA and HCC, and they vary among patients’ subgroups. Interestingly, tumor resection induces normalization of CD8^+^ Tregs’ percentage (but not absolute counts) in CCA, and, to a much lesser extent, in HCC. These alterations point to an active role of CD8^+^ Tregs in CCA and HCC physiopathology, and future studies focusing on CD8^+^ Tregs levels and function, analyzing in parallel the peripheral blood, peritumoral, and tumoral tissues, may be valuable to the discovery of new biomarkers and prognostic factors of these diseases.

Regarding circulating lymphocytes, a great interindividual heterogeneity in the percentages of the distinct T cell subsets from different cancer patients was observed. We report a decreased percentage of circulating CD4^+^ T cells, as previously described by others, in HCC [20]. Notably, we detected a decrease in the frequency of circulating Th1/17 cells (CD4^+^ T lymphocytes with the plasticity to produce both IFN-γ and IL-17 cytokines), together with a decrease in the frequency of circulating Tc17 cells in CCA and HCC patients, pointing to an active migration of Th1/17 and Tc17 cells from the peripheral blood to the tumor microenvironment. In line with this, a recent study showed an increased proportion of intratumoral Tc17 cells, compared to the peripheral blood, in HCC [41]. Importantly, Tc17 cells infiltrating the tumor have been demonstrated to possess protumoral activity, and to express high levels of CCL20, thereby recruiting Tregs [41]. These facts are in accordance with our previous research, carried out in this same cohort of patients, describing tumor infiltration by Th1/17 and Tc17 cells and elevated circulating levels of CCL20 [22,23]. Remarkably, higher levels of tumor-infiltrated Tc17 cells have been associated with an unfavorable clinical course in HCC [41].

A better understanding of the potential that the inflammatory response may have on the inhibition of the tumor progression can foster the development of new and improved immunomodulatory approaches for the treatment of liver carcinomas. Being aware that tumor resection and liver transplants are the most applied treatments for liver carcinomas [42,43], it is important to assess the effect of these therapies on the homeostasis of the immune system in liver cancer patients.

Our study has some limitations, namely the small sample size and the lack of a long-term follow-up to assess the persistence of immune recovery post-tumor-resection. The limited sample size, especially for CCA, undeniably limits the analysis of patients’ subgroups (TNM stage, histologic grade, presence vs. absence of cirrhosis, microvascular invasion, aggressiveness degree, relapsing vs. recurrence-free CCA and HCC) and the conclusions on a possible value of circulating immune cells as biomarkers for prognosis and/or disease aggressiveness. The small sample size also restricts the discovery of potential interindividual variability in Treg response due to differences in patients’ immune status (especially for HCC vs. CCA). Despite these shortcomings, our results open new avenues of research, focusing on specific immune cell populations in larger cohorts of patients, in order to establish their value as prognostic biomarkers, providing a good starting point for new studies in this field.

## 5. Conclusions

In conclusion, our results point to alterations in the immune systems of CCA and HCC patients that converge in a decreased ability to develop a proinflammatory response, observed here by the diminished percentage of circulating T cells expressing IFN-γ and/or IL-17. Interestingly, the surgical resection of the tumor induces changes in the immune system that are detected at the peripheral blood level. This study demonstrated that tumor resection partially restores circulating CD4^+^ Treg cell levels in CCA and HCC patients. These cells constitute an important resource to maintain the immune system homeostasis by regulating the innate and acquired immune response and shaping inflammation and its resolution. However, further studies with larger sample sizes and long-term follow-up are needed to validate the findings presented here.

## Figures and Tables

**Figure 1 jcm-13-06011-f001:**
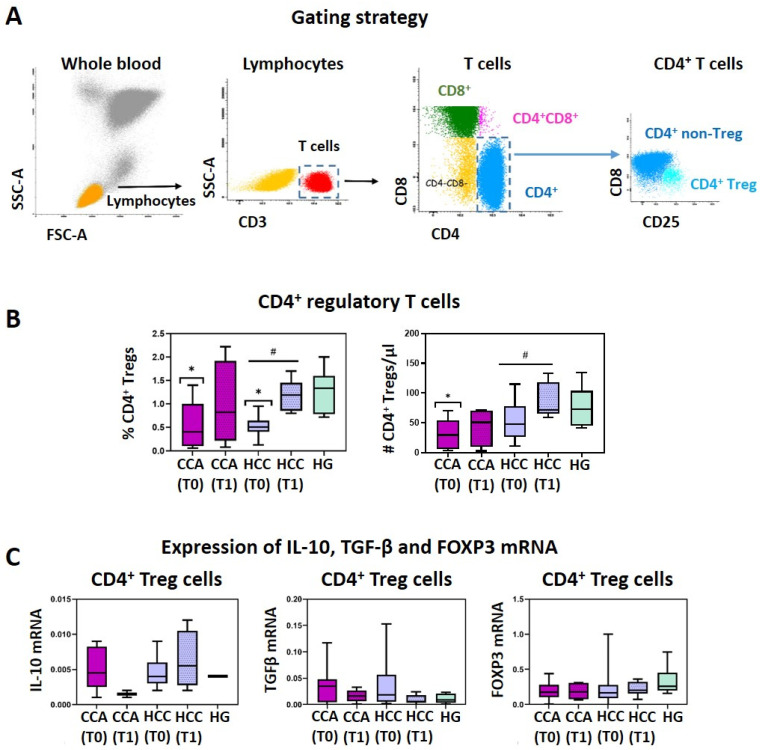
Characterization of circulating CD4^+^ T regulatory cells. (**A**) Bivariate dot plot histograms illustrating the identification of circulating lymphocytes (orange events), T lymphocytes (red events) and their cell subsets (CD4^+^ T cells are represented in blue, CD8^+^ T cells in green, CD4^+^ CD8^+^ T cells in pink, and CD4^−^CD8^−^ T cells are represented in yellow), and CD4^+^ Treg cells (light blue events), by flow cytometry. (**B**) Boxplots with the frequency (%) of circulating CD4^+^ T regulatory cells (CD4^+^ Tregs, measured in whole blood) and their absolute numbers (cells/μL) in cholangiocarcinoma (CCA) and hepatocellular carcinoma (HCC) patients, at T0 and T1, and in the healthy group (HG). (**C**) Boxplots with the mRNA expression levels of IL-10, TGF-β, and FOXP3 among CD4^+^ Tregs purified from CCA and HCC patients, at T0 and T1, and from the HG. Statistically significant differences were considered when *p* < 0.05: * between the groups indicated in the figure and the HG; # between T0 and T1.

**Figure 2 jcm-13-06011-f002:**
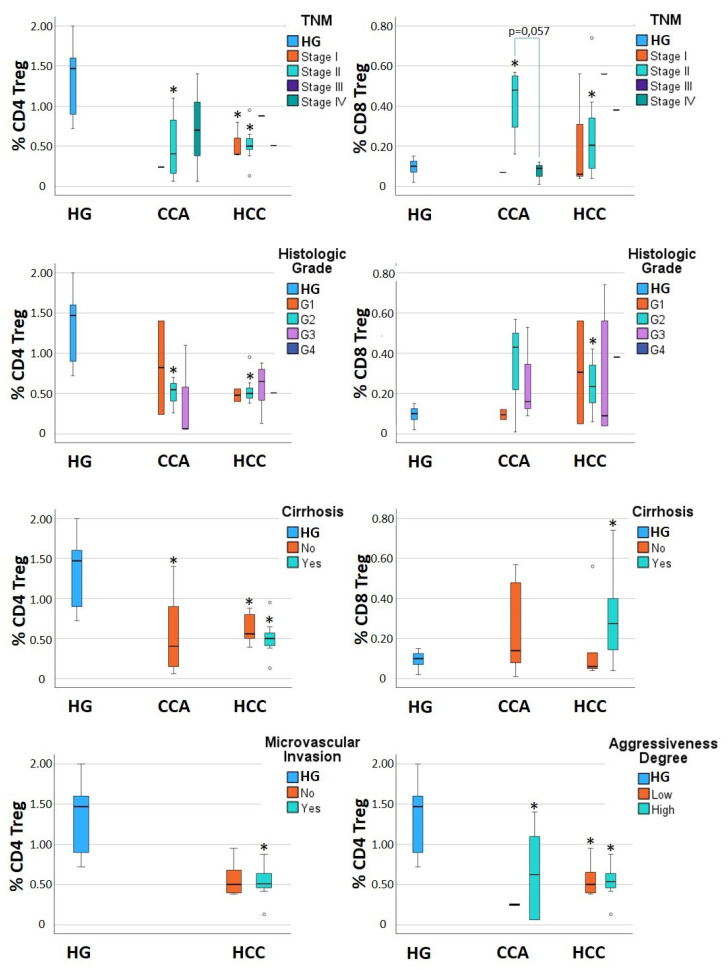
Boxplots displaying the percentage of circulating CD4^+^ T regulatory cells (CD4 Treg), measured in whole blood, and CD8^+^ T regulatory cells (CD8 Treg), measured within CD8^+^ T cells, in heathy individuals (HG) and patients with cholangiocarcinoma (CCA) and hepatocellular carcinoma (HCC). The percentages of Treg cells were analyzed before tumor resection (at T0), and the CCA and HCC patients were grouped according to TNM stage, histologic grade, the presence or absence of liver cirrhosis and microvascular invasion, and the aggressiveness degree of the disease. Statistically significant differences were considered when *p* < 0.05: * between the groups indicated in the figure and the HG.

**Table 1 jcm-13-06011-t001:** Clinical data from cholangiocarcinoma (CCA) and hepatocellular carcinoma (HCC) patients enrolled in this study. Number of patients and frequencies (%) are indicated.

		CCA n = 8	HCC n = 20
TNM	Stage I	1 (13%)	3 (15%)
	Stage II	4 (50%)	15 (75%)
	Stage III	0 (0%)	1 (5%)
	Stage IV	3 (38%)	1 (5%)
Histologic grade	G1	2 (25%)	2 (10%)
	G2	3 (38%)	11 (55%)
	G3	3 (38%)	6 (30%)
	G4	0 (0%)	1 (5%)
HBsAg	Positive	0 (0%)	1 (5%)
HCV	Positive	0 (0%)	6 (30%)
Microvascular invasion	Positive	-	8 (40%)
Cirrhosis	Positive	0 (0%)	15 (75%)
Disease aggressiveness	Low degree	2 (25%)	10 (50%)
	High degree	6 (75%)	10 (50%)
Treatment	Liver transplant	0 (0%)	7 (35%)
	Tumor resection	8 (100%)	13 (65%)

TNM, Tumor, node, and metastasis staging; HBsAg, hepatitis B surface antigen; HCV, hepatitis C virus.

**Table 2 jcm-13-06011-t002:** Panel of monoclonal antibody reagents (with clones and commercial source) used for the immunophenotypic characterization of NK and T cells, and for Treg cell purification by fluorescence-activated cell sorting.

	V450	V500-c	FITC	PE	PE-Cy7	APC	APC-H7
**Tube 1**	CD4	CD45	CD127	CD25		CD3	CD8
Clone	RPA-T4	2D1	HIL-7R-M21	B1.49.9		SK7	SK1
Commercial source	BD	BD	BD Pharmingen	Beckman Coulter		BD	BD
**Tube 2**	CD4	CD45	cyIFN-γ	cyIL-17	CD56	CD3	CD8
Clone	RPA-T4	2D1	4S.B3	SCPL1362	N901	SK7	SK1
Commercial source	BD	BD	BD Pharmingen	BD Pharmingen	Beckman Coulter	BD	BD

Commercial sources: BD (Becton Dickinson Biosciences (BD), San Jose, CA, USA), BD Pharmingen (San Diego, CA, USA), Beckman Coulter (Miami, FL, USA). APC, allophycocyanin; APC-H7, allophycocyanin-hilite 7; cy, cytoplasmic staining; FITC, fluorescein isothiocyanate; IFN-γ, interferon gamma, IL-17, interleukin 17; PE, phycoerythrin; PE-Cy7, phycoerythrin-cyanine 7.

**Table 3 jcm-13-06011-t003:** Frequency of peripheral blood CD4^+^ T regulatory cells (CD4 Treg), measured in whole blood) and CD8^+^ T regulatory cells (CD8 Treg), measured within CD8^+^ T cells, in heathy individuals (HG), in patients with cholangiocarcinoma (CCA) and hepatocellular carcinoma (HCC), and their subgroups, analyzed before tumor resection (at T0).

	CCA		HCC		HG	
	CD4 Treg Mean ± SD	CD8 Treg Mean ± SD	CD4 Treg Mean ± SD	CD8 Treg Mean ± SD	CD4 Treg Mean ± SD	CD8 Treg Mean ± SD
**Total**	**0.55 ± 0.49 ***	0.25 ± 0.22	**0.71 ± 0.54 ***	0.37 ± 0.48	1.30 ± 0.44	0.16 ± 0.21
**TNM**						
**Stage I**	0.24 ± NA	0.07 ± NA	0.53 ± 0.23 *	0.22 ± 0.29		
**Stage II**	**0.49 ± 0.45 ***	**0.42 ± 0.18 ***	**0.76 ± 0.62 ***	0.38 ± 0.54		
**Stage III**	NA	NA	0.88 ± NA	0.56 ± NA		
**Stage IV**	0.72 ± 0.67	0.07 ± 0.06	0.51 ± NA	0.38 ± NA		
**Histologic Grade**						
**G1**	0.82 ± 0.82	0.10 ± 0.04	0.48 ± 0.11	0.31 ± 0.36		
**G2**	**0.50 ± 0.22 ***	0.34 ± 0.29	0.67 ± 0.41	**0.44 ± 0.62 ***		
**G3**	0.41 ± 0.60	0.26 ± 0.24	0.89 ± 0.81	0.26 ± 0.31		
**G4**	NA	NA	0.51 ± NA	0.38 ± NA		
**Liver Cirrhosis**						
**No**	0.55 ± 0.49	0.25 ± 0.22	**0.63 ± 0.21 ***	0.17 ± 0.22		
**Yes**	NA	NA	**0.74 ± 0.63 ***	**0.44 ± 0.43 ***		
**Microvascular Invasion**						
**No**	NA	NA	0.86 ± 0.68	0.28 ± 0.23		
**Yes**	NA	NA	0.53 ± 0.23 *	0.49 ± 0.73		
**Aggressiveness Degree**						
**Low**	0.25 ± 0.01 *	0.32 ± 0.35	0.68 ± 0.42*	0.24 ± 0.17		
**High**	0.65 ± 0.54 *	0.22 ± 0.21	0.75 ± 0.67 *	0.49 ± 0.65		

CCA and HCC patients were grouped according to TNM stage, histologic grade, the presence or absence of liver cirrhosis and microvascular invasion, and the aggressiveness degree of the disease. NA, not analyzed. Statistically significant differences were considered when *p* < 0.05: * between the groups indicated in the table and the HG, using the Mann–Whitney U-test. The results are given by mean ± standard deviation. Mean ± SD in bold whenever *p* < 0.05 vs. HG.

**Table 4 jcm-13-06011-t004:** Frequency (%) and absolute numbers ([], number of cells/μL) of peripheral blood CD4^+^ T regulatory cells (CD4 Treg) and CD8^+^ T regulatory cells (CD8 Treg) from cholangiocarcinoma (CCA) patients, subdivided in patients with recurrent CCA and recurrence-free patients after a 12 month follow-up, at the time of the surgical procedure (T0), and once the patients were recovered from surgery (T1), and in a group of healthy individuals (HG).

	Recurrent CCA		Recurrence-Free CCA		HG
	T0 n = 3	T1 n = 2	T0 n = 4	T1 n = 2	n = 10
	Mean ± SD	Mean ± SD	Mean ± SD	Mean ± SD	Mean ± SD
**% CD4+ Treg cells** **(within leukocytes)**	0.51 ± 0.77	0.55 ± 0.66	**0.65 ± 0.36 ^a^**	1.43 ± 1.12	1.30 ± 0.44
**[CD4+ Treg cells/µL]**	26 ± 38	66 ± 7	40 ± 18	75 ± 60	77 ± 33
**% CD8+ Treg cells** **(within CD8+ T cells)**	0.12 ± 0.04	NA	0.26 ± 0.26	0.17 ± 0.16	0.16 ± 0.21
**% CD8+ Treg cells** **(within leukocytes)**	0.0063 ± 0.0052	NA	0.0166 ± 0.0216	0.0112 ± 0.0061	0.0094 ± 0.0048
**[CD8+ Treg cells/µL]**	0.36 ± 0.24	NA	1.00 ± 1.14	0.51 ± 0.40	0.59 ± 0.25

A Mann–Whitney U-test was performed to compare each group of patients vs. the healthy group (^a^), with a significance level of 0.05 (*p* < 0.05). A Wilcoxon test was performed to compare T1 vs. T0 (^b^), with a significance level of 0.05 (*p* < 0.05). The results are given by mean ± standard deviation (SD). Mean ± SD in bold whenever *p* < 0.05 vs. HG.

**Table 5 jcm-13-06011-t005:** Frequency (%) and absolute numbers ([], number of cells/μL) of peripheral blood lymphocyte subsets in cholangiocarcinoma (CCA) and hepatocellular carcinoma (HCC) patients, at the time of the surgical procedure (T0), once the patients were recovered from surgery (T1), and in a group of healthy individuals (HG). The protein expression levels of IFN-γ and IL-17 (measured as mean fluorescence intensity, MFI) are also indicated.

		CCA n = 8		HCC n = 20		HG n = 10
		T0	T1	T0	T1	
		Mean ± SD	Mean ± SD	Mean ± SD	Mean ± SD	Mean ± SD
**% Lymphocytes**		33 ± 14	37 ± 12	29 ± 14	28 ± 9	42 ± 8
**[Lymphocytes/µL]**		**1903 ± 585 ^a^**	1253 ± 127	2730 ± 2798	2220 ± 904	2392 ± 500
**% T cells**		22 ± 12	27 ± 11	20 ± 9	21 ± 9	29 ± 8
**[T cells/µL]**		1305 ± 520	1281 ± 882	1619 ± 977	1631 ± 802	1539 ± 395
**% CD4^+^ T cells**		57 ± 19	59 ± 16	61 ± 16	63 ± 15	66 ± 10
	% IFN-γ^+^	15 ± 9	19 ± 6	23 ± 15	20 ± 14	19 ± 8
	MFI IFN-γ^+^	6667 ± 2294	8862 ± 2543	6887 ± 3206	7286 ± 3823	8531 ± 1983
	% IL-17^+^	0.90 ± 0.51	1.45 ± 0.94	1.42 ± 1.15	**1.14 ± 0.82 ^b^**	1.63 ± 1.45
	MFI IL-17^+^	557 ± 110	619 ± 131	621 ± 173	514 ± 159	534 ± 124
	% IFN-γ^+^IL-17^+^	**0.11 ± 0.14 ^a^**	0.15 ± 0.11	0.21 ± 0.28	**0.16 ± 0.11 ^b^**	0.29 ± 0.22
**% CD8^+^ T cells**		39 ± 20	37 ± 17	32 ± 17	32 ± 16	28 ± 8
	% IFN-γ^+^	43 ± 23	59 ± 19	52 ± 26	47 ± 35	47 ± 13
	MFI IFN-γ^+^	5512 ± 2714	7189 ± 2313	5280 ± 3061	5487 ± 2886	6630 ± 1836
	% IL-17^+^	**0.12 ± 0.17 ^a^**	**0.08 ± 0.04 ^a^**	0.15 ± 0.19	**0.17 ± 0.15 ^b^**	0.20 ± 0.12
	MFI IL-17^+^	365 ± 162	540 ± 211	470 ± 152	385 ± 118	438 ± 190
	% IFN-γ^+^IL-17^+^	0.08 ± 0.15	0.03 ± 0.02	0.08 ± 0.013	0.06 ± 0.06	0.09 ± 0.06
	% CD56^+^	12 ± 6.7	9 ± 5	16 ± 14	17 ± 14	10 ± 9
**% CD4^+^CD8^+^ T cells**		1.5 ± 0.8	1.2 ± 0.26	2.2 ± 1.0	1.5 ± 2.1	1.0 ± 0.8
	% IFN-γ^+^	35 ± 25	44 ± 15	44 ± 29	32 ± 19	41 ± 16
	MFI IFN-γ^+^	5545 ± 2485	6513 ± 2684	6049 ± 3258	5660 ± 3172	6941 ± 1858
	% CD56^+^	13 ± 14	12 ± 6	14 ± 11	11 ± 8	12 ± 9

The percentages of lymphocytes and T cells were measured within leukocytes; the percentages of CD4^+^ T cells, CD8^+^ T cells, and CD4^+^CD8^+^ T cells were measured within T cells. A Mann–Whitney U-test was performed to compare each group of patients vs. the healthy group (^a^), with a significance level of 0.05 (*p* < 0.05). A Wilcoxon test was performed to compare T1 vs. T0 (^b^), with a significance level of 0.05 (*p* < 0.05). The results are given by mean ± standard deviation (SD). Mean ± SD in bold whenever *p* < 0.05 vs. HG.

**Table 6 jcm-13-06011-t006:** Frequency (%) of peripheral blood natural killer (NK) cell subsets present in cholangiocarcinoma (CCA) and hepatocellular carcinoma (HCC) patients, at the time of surgical procedure (T0), once the patients were recovered from surgery (T1), and in a group of healthy individuals (HG). Protein expression levels of IFN-γ and IL-17 (measured as mean fluorescence intensity, MFI) are also indicated.

		CCA n = 8		HCC n = 20		HG n = 10
		T0	T1	T0	T1	
		Mean ± SD	Mean ± SD	Mean ± SD	Mean ± SD	Mean ± SD
**% NK cells**		2.6 ± 2.6	1.7 ± 0.9	3.2 ± 2.3	3.0 ± 1.7	2.3 ± 1.0
**% CD56^DIM^ NK cells**		82 ± 13	83 ± 12	91 ± 6	92 ± 4	88 ± 5
	% IFNγ^+^	30 ± 27	54 ± 26	39 ± 29	35 ± 27	42 ± 25
	MFI IFNγ	1705 ± 766	2063 ± 888	1704 ± 761	1694 ± 583	1913 ± 604
	% CD8^+^	4.1 ± 3.7	6.4 ± 4.0	3.4 ± 3.4	3.1 ± 3.8	2.3 ± 2.2
	HLADR^+^	2.3 ± 1.6	2.1 ± 1.9	1.2 ± 1.5	0.9 ± 0.8	2.0 ± 1.8
**% CD56^BRIGHT^ NK cells**		17 ± 13	17 ± 13	8.7 ± 5.3	7.8 ± 4.4	12 ± 5
	% IFNγ^+^	26 ± 29	34 ± 21	38 ± 30	23 ± 21	28 ± 14
	MFI IFNγ	2157 ± 952	1892 ± 997	1916 ± 1054	1805 ± 711	1939 ± 604
	% CD8^+^	5.6 ± 10	4.0 ± 2.9	3.8 ± 4.1	**5.8 ± 5.7 ^a^**	1.3 ± 1.3

The percentage of NK cells was measured within leukocytes. The percentages of CD56^DIM^ and CD56^BRIGHT^ NK cell subsets were measured within NK cells. An independent-samples Mann–Whitney U-test was performed to compare each group of patients vs. the healthy group (^a^), with a significance level of 0.05 (*p* < 0.05). A Wilcoxon test was performed to compare T1 vs. T0 (^b^), with a significance level of 0.05 (*p* < 0.05). The results are given by mean ± standard deviation (SD). Mean ± SD in bold whenever *p* < 0.05 vs. HG.

## Data Availability

The data presented in this study are available in this article, including in the Appendix A.

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
