# Peer review of "Tumor Resection in Hepatic Carcinomas Restores Circulating T Regulatory Cells"

_jcm, 2024, doi:10.3390/jcm13196011_

Round 1

Reviewer 1 Report

Comments and Suggestions for Authors

Q1:Introduction: some sections are a bit of redundancy when discussing the immunosuppressive role of Tregs.

Q2: Materials and Methods:  Please explain detailedlyf how flow cytometry gating strategies were validated. The methods for cytokine production could use additional clarity regarding controls used for comparison.

Q3: The statistical analysis appears sound, but certain tests (e.g., Mann-Whitney U test) should be emphasized as non-parametric due to the small sample size. The use of appropriate statistical software like SPSS is mentioned, which is positive.

Q4: lines 98–100: The decision to exclude liver transplant patients  from T1 analysis is important but might be missed. A clearer explanation of the reasons for exclusion should be included here (e.g., "To ensure that immunosuppressive effects of post-transplant drugs like tacrolimus did not interfere with the natural restoration of Treg cells…").

Q5: Table 2 : Please address the antibodies used here were validated or cross-reacted for specific cell markers.

Q6:Please explain why 20 HCC and 8 CCA patients were selected and whether this number provides enough statistical power.

Q7: Result: there is an inconsistency in formatting regarding the spacing before numbers, for example, "p > 0.05" versus "p< 0.05." Consistency is important for readability.

Q8: When discussing figures, such as differences in Treg counts, more detailed explanations of the significance of these differences could improve readability. For example, instead of just stating "no significant differences," explaining the biological relevance of the non-significance can offer more context

Q9: Results:Adding a summary table to highlight key findings in subgroup comparisons might help readers absorb the main points more easily.

Q10: Results:  Figure 1: “The frequency of circulating CD4+ Tregs is significantly reduced in CCA and HCC patients before tumor resection.”Consider : “Figure 1B demonstrates a significant reduction in circulating CD4+ Treg levels in both CCA and HCC patients at T0 compared to healthy controls (p < 0.05), with levels recovering post-surgery (T1). This suggests a potential migration of Tregs into the tumor microenvironment pre-surgery.”

Q11: Discussion: The introduction of new ideas (like MEOX1 and its association with Treg function) that seem disconnected from the earlier narrative.

Q12:Discussion: Streamline the discussion by focusing on how your findings contribute new knowledge to the field, particularly the restoration of Tregs after tumor resection.

Q13: Some points, such as the importance of CD4+ Tregs in cancer, are reiterated multiple times.

Q14:Please consider to address the limitations including: 1.The small sample size's effect on the statistical power of the findings. 2. The potential variability in Treg response due to differences in individual patient immune status (especially for HCC vs CCA). 3. Lack of long-term follow-up to assess the persistence of immune recovery post-resection.

Q15: Conclusion: The conclusion does a good job of summarizing the findings but could be made stronger by emphasizing the clinical relevance of restoring Treg levels.

Q16: Reference:The references in the paper often include periods after abbreviations inconsistently, e.g., "FoxP3" and "Foxp3." The journal may have a specific convention on how gene names should be styled.

Q17:line 355:the immune system homeostasis induced by tumor resection  could be phrased more clearly as "restoration of immune system balance following tumor resection."

Q18: Supplementary tables (S1) are referenced without much context in the text, which could leave readers unsure of where to find the relevant data.

Q19: line 47-48: "The frequency of peripheral CD4+ Treg cells was restored after tumor resection." I would suggest you to rephrase as following: "The frequency of peripheral CD4+ Treg cells increased from X% at T0 to Y% at T1 following tumor resection."

Comments on the Quality of English Language

Minor editing of English language required.

Reviewer 2 Report

Comments and Suggestions for Authors

The authors provide a very good manuscript with solid data, presented in a very professional manner. I only have a few, minor, corrections/suggestions. More specifically:

1. The introduction section, provides enough data regarding tumor 'immune-environment', especially when it comes to lymphocytes. The authors should add references regarding Tregs in non-liver tumors (ref: doi: 10.1186/s12943-020-01234-1; doi: 10.2174/1570162X18666200401122922; doi: 10.1038/s41467-023-44391-9). Moreover, the last line, providing the study results should be deleted from the introduction.

3. The materials and methods and results sections are well written and provide all needed databoth in order to re-do the experiment as well as understanding the results.

4. The discussion section is also well written with all needed data. However, a little limitations paragraph should be added at the end of the manuscript.

Comments on the Quality of English Language

English needing minor editing

Round 2

Reviewer 1 Report

Comments and Suggestions for Authors

It seems that the authors have made a great effort in revising their manuscript.  The corrected version is good. I am pleased with it. 

Author Response

Thank you very much for thoroughly revised our manuscript. 

It definitely contributed to improve our paper.